# Interim safety and efficacy of gene therapy for *RLBP1*-associated retinal dystrophy: a phase 1/2 trial

Anders Kvanta [1] ✉, Nalini Rangaswamy[2], Karen Holopigian[3], Christine Watters[4], Nicki Jennings[2], Melissa S. H. Liew[2], Chad Bigelow[2], Cynthia Grosskreutz[2], Marie Burstedt[5], Abinaya Venkataraman[1], Sofie Westman[1], Asbjörg Geirsdottir[1], Kalliopi Stasi[2] & Helder André [1]

Gene therapy holds promise for treatment of inherited retinal dystrophies, a group of rare genetic disorders characterized by severe loss of vision. Here, we report up to 3-year pre-specified interim safety and efficacy results of an open-label first-in-human dose-escalation phase 1/2 gene therapy clinical trial in 12 patients with retinal dystrophy caused by biallelic mutations in the retinaldehyde-binding protein 1 (*RLBP1*) gene of the visual cycle. The primary endpoints were systemic and ocular safety and recovery of dark adaptation. Secondary endpoints included microperimetry, visual field sensitivity, dominant eye test and patient-reported outcomes. Subretinal delivery of an adeno-associated viral vector (AAV8-*RLBP1*) was well tolerated with dose-dependent intraocular inflammation which responded to corticosteroid treatment, and focal atrophy of the retinal pigment epithelium as the dose limiting toxicity. Dark adaptation kinetics, the primary efficacy endpoint, improved significantly in all dose-cohorts. Treatment with AAV8-*RLBP1* resulted in the resolution of disease-related retinal deposits, suggestive of successful restoration of the visual cycle. In conclusion, to date, AAV8-*RLBP1* has shown preliminary safety and efficacy in patients with *RLBP1*-associated retinal dystrophy. Trial number: NCT03374657.

Retinal dystrophy (RD) caused by biallelic mutations in the Retinaldehyde-binding protein 1 (*RLBP1*) gene, is an autosomal recessive rod-cone dystrophy that affects approximately 10,000 individuals worldwide[1,2]. This condition, for which there is currently no treatment, belongs to the retinitis pigmentosa (RP) family of inherited retinal degenerations (IRDs) and is characterized by retinal white dot-like deposits (punctata albescens) in earlier stages and patches of chorioretinal atrophy in later stages of the disease. Night blindness and delayed dark adaptation are observed in childhood, with eventual loss of mid-peripheral visual field sensitivity and central vision. Disease progression is slow, with a decline in visual acuity typically beginning in early adulthood and legal blindness predominantly occurring in middle age[3,4]. Bothnia dystrophy and Newfoundland rod-cone dystrophy are two severe forms of *RLBP1*-RD distinguished by early atrophy of the macula, with the former particularly prevalent in northern Sweden, where affected patients typically carry a c.700 C > T mutation[3–5].

A natural history study followed 44 patients with Bothnia- and Newfoundland-type *RLBP1*-RD for 5 years and explored various potential clinical endpoints for this phase 1/2 clinical trial. Based on the

[1]Department of Clinical Neuroscience, St. Erik Eye Hospital, Karolinska Institutet, Stockholm, Sweden. [2]Novartis Institutes for Biomedical Research, Cambridge, MA, USA. [3]Novartis Institutes for Biomedical Research, East Hanover, NJ, USA. [4]Novartis Pharmaceuticals Corporation, Cambridge, MA, USA. [5]Department of Clinical Sciences/Ophthalmology, University of Umeå, Umeå, Sweden. ✉e-mail: anders.kvanta@ki.se

**Fig. 1 | Diagram of patient flow.** This figure shows reasons for exclusion from the study and the number of patients included in the analyses.

characteristic symptoms of severely delayed dark adaptation recovery in this *RLBP1*-RD patient population, a prototype custom-designed 6-hour long dark adaptation kinetics test was evaluated in this natural history study. It showed severely delayed dark adaptation sensitivity recovery in all patients across all age groups (17–69 years), making it a potentially suitable primary efficacy outcome for this specific patient population[4].

The *RLBP1* gene is expressed in the retinal pigment epithelium (RPE) and Müller cells and encodes for cellular retinaldehyde-binding protein (CRALBP). CRALBP binds to 11-*cis*-retinol in the visual cycle and thereby augments the activity of retinoid isomerohydrolase RPE65 and facilitates oxidation of 11-*cis*-retinol to 11-*cis*-retinal[6]. An important landmark was reached when gene therapy for Leber's congenital amaurosis (*RPE65*-associated RD) was approved, and further paved the way for potential treatment of other IRDs[7]. Gene therapy with adeno-associated viral (AAV) vectors expressing a corrected *RLBP1* gene (AAV8-*RLBP1*) restored CRALBP levels and improved dark adaptation in *Rlbp1*-deficient mice[8,9]. Based on the successful results from AAV8-*RLBP1* treatment in *Rlbp1*-deficient mice and following the establishment of safety in non-human primates (NHPs)[8,10], we assessed the safety, tolerability, and efficacy of AAV8-*RLBP1* in a phase 1/2 first-in-human dose escalation trial in patients with *RLBP1*-RD (https://clinicaltrials.gov/ct2/show/NCT03374657, https://www.clinicaltrialsregister.eu/ctr-search/trial/2016-002696-10/SE).

## Results

### Demographics and surgical procedure
A total of 13 patients were screened for eligibility, including confirmation of biallelic mutation in the *RLBP1* gene (Fig. 1, Table 1, Table 2). Disease phenotypes included in the trial ranged from advanced stages with widespread chorioretinal atrophy (which by design were enrolled in the first cohort for enhanced risk profile) to earlier disease progression (such as retinitis punctata albescens). Three of the patients treated in this trial had characteristic punctata albescens ("white dots") fundus appearance (Table 1). One patient was diagnosed with stomach cancer shortly after screening (and before dosing with AAV8-*RLBP1*) and was screen-failed. The remaining 12 patients (C1.A to C4.C) were enrolled into four dose-escalating cohorts (C1 to C4) and treated with a single subretinal injection of AAV8-*RLBP1* (Table 1, Table 2 and Supplementary Fig. 1). These patients consisted of 4 male and 8 female Caucasians, aged 31 to 66 years, with a best corrected visual acuity (BCVA) ranging from light perception to 64 letters. All patients were treated in the non-dominant eye (poorer seeing eye), except for patient C2.C who met all inclusion criteria only in the dominant eye. All surgeries were performed without complications, and subretinal delivery of AAV8-*RLBP1* vector rendered blebs

that predominantly formed centrally extending outside the macula (in 7 patients) or peripherally (in 5 patients). All patients were dosed with the pre-determined dose and volume (see Table 2). The bleb successfully covered the fovea in 11 patients (Supplementary Fig. 2). In patient C3.B, a smaller-than-expected bleb was formed, and in C3.C, the bleb formed eccentrically, covering only the superior peripheral retina.

### Safety and tolerability
During the one to three years of follow-up, a total of 108 adverse events (AEs) were reported – 65 ocular and 43 non-ocular – in all cohorts; most were mild and unrelated to the study treatment (Supplementary Table 1). Four severe adverse events (SAEs) were reported, 2 non-ocular and 2 ocular. The non-ocular SAEs were fractures, of the pelvis and arm respectively, in patients C1.A and C1.B. Both SAEs were unrelated to the study drug or procedure and resolved with appropriate treatment. Two ocular SAEs, one procedure-related that resolved after appropriate treatment and one study-drug related that resolved partially without specific treatment, occurred in patients C1.C and C4.C, respectively. A procedure-related SAE of elevated intraocular pressure (IOP) of 43 mmHg, was noted 1 day post-operatively in patient C1.C. This was due to zonular instability from the air tamponade, which induced forward displacement of the intraocular lens that led to angle closure. When the IOP did not normalize with medication, surgical evacuation of air successfully restored IOP within the normal range on post-treatment day 3. The second ocular SAE occurred in patient C4.C; a severe loss of vision of 59 letters (baseline BCVA of 64 letters decreased to 5 letters at 2 months post-treatment, was 13 letters at year 1). No other ocular findings were noted for this patient post-treatment. Subsequent follow-ups with fluorescein angiography, MRI, CT scan, and carotid Doppler were all normal. At the 1-year follow-up, the vision had partially recovered both objectively and subjectively (Supplementary Fig. 3 and Supplementary Table 2).

Since intraocular inflammation was a dose-limiting finding in the NHP toxicology study, all patients received prophylactic corticosteroids per protocol in a regimen similar to previous subretinal gene therapy trials[10,11]. The regimen was prolonged for patients in later cohorts (details provided in Methods). For two patients (C2.B and C2.C), inferior subretinal exudates with focal serous retinal detachment were noted shortly after surgery. These were considered study drug-related and resolved or regressed within 3 months without adjustment of the steroid regimen (Fig. 2A, Table 3, Supplementary Fig. 4A). Intraocular inflammation was observed in two patients in C3 and in one patient in C4 and was considered study drug-related (Table 3).

Patient C3.A complained of blurred vision 3 weeks after AAV8-*RLBP1* dosing due to iritis with mild vitritis that resolved after 2 weeks and 2 months, respectively, with oral prednisolone 40 mg daily tapered over 4 months and dexamethasone eyedrops 8 times daily tapered over 6 weeks. Patient C3.C showed mild vitritis 4 weeks post-AAV8-*RLBP1* dosing that resolved with oral prednisolone 40 mg daily tapered over 4 months and dexamethasone eyedrops 3 times daily tapered over 6 weeks. Based on these adverse events, the data monitoring committee (DMC) recommended prolonged oral corticosteroid tapering over 12 weeks for C4 (details provided in "Methods"). Patient C4.C showed mild vitritis with unexpected visual loss 2 weeks post- AAV8-*RLBP1* dosing (before the initiation of the planned tapering of oral prednisolone prophylaxis), which resolved after 2 weeks of preplanned oral prednisolone 40 mg daily. Due to the unexpected vision loss in this patient, 4 weeks after dosing, prednisolone was increased to 60 mg daily for 3 weeks before tapering off for 10 weeks.

Pigmented subretinal deposits adjacent to the retinotomy site, considered to be study drug-related, occurred within a month after treatment in one patient in C3 (C3.B) and two patients in C4 (C4.A and C4.C). These were visible as hyperfluorescent dots on *en face* fundus

**Table 1 | Patient demographics and phenotype**

| Cohort | ID# | Treatment Year | Baseline BCVA Treated eye ETDRS (logMar) | Baseline BCVA Untreated eye ETDRS (logMar) | Stage[a] | Punctata albescens[b] | Follow-up (yrs) |
|---|---|---|---|---|---|---|---|
| C1 | C1.A | 2018 | LP (2.6) | LP (2.6) | Late | No | 3 |
| | C1.B | 2018 | HM (2.3) | 7 (1.6) | Late | No | 3 |
| | C1.C | 2019 | 26 (1.2) | 43 (0.8) | Late | No | 3 |
| C2 | C2.A | 2019 | 15 (1.4) | 21 (1.3) | Early | No | 3 |
| | C2.B | 2019 | 40 (0.9) | 39 (0.9) | Early | Yes | 3 |
| | C2.C | 2019 | 28 (1.1) | 24 (1.2) | Intermediate | Yes | 3 |
| C3 | C3.A | 2020 | 28 (1.1) | 27 (1.2) | Late | No | 2 |
| | C3.B | 2020 | 36 (1.0) | 59 (0.5) | Late | No | 2 |
| | C3.C | 2020 | 28 (1.1) | 37 (1.0) | Early | Yes | 2 |
| C4 | C4.A | 2020 | 27 (1.2) | 31 (1.1) | Intermediate | No | 2 |
| | C4.B | 2021 | 16 (1.4) | 15 (1.4) | Intermediate | No | 1 |
| | C4.C | 2021 | 64 (0.4) | 66 (0.4) | Late | No | 1 |

[a]Disease stage defined by the total area of chorioretinal fundus atrophy: early = no atrophy, intermediate = ≤50% atrophy, late = >50% atrophy.
[b]Presence of white puncta scattered around the vascular arcades.
*LP* light perception, *HM* hand motion, *BCVA* best-corrected visual acuity, *ETDRS* early treatment diabetes retinopathy study letters.

**Table 2 | Genotypes and dosing of AAV8-*RLBP1***

| Cohort | ID# | Mutation (allele 1/2) | Volume injected (microliter) | Vector concentration (vg/microliter) | Vector dose (vg/eye) |
|---|---|---|---|---|---|
| C1 | C1.A | 677 T > A/700 C > T | 150 | $3.3 \times 10^7$ | $5 \times 10^9$ |
| | C1.B | 677 T > A/700 C > T | 150 | $3.3 \times 10^7$ | $5 \times 10^9$ |
| | C1.C | 700 C > T/700 C > T | 150 | $3.3 \times 10^7$ | $5 \times 10^9$ |
| C2 | C2.A | 700 C > T/700 C > T | 300 | $3.3 \times 10^7$ | $1 \times 10^{10}$ |
| | C2.B | 700 C > T/700 C > T | 300 | $3.3 \times 10^7$ | $1 \times 10^{10}$ |
| | C2.C | 700 C > T/700 C > T | 300 | $3.3 \times 10^7$ | $1 \times 10^{10}$ |
| C3 | C3.A | 700 C > T/700 C > T | 300 | $1 \times 10^8$ | $3 \times 10^{10}$ |
| | C3.B | 700 C > T/700 C > T | 300 | $1 \times 10^8$ | $3 \times 10^{10}$ |
| | C3.C | 700 C > T/700 C > T | 300 | $1 \times 10^8$ | $3 \times 10^{10}$ |
| C4 | C4.A | 700 C > T/700 C > T | 300 | $3.3 \times 10^8$ | $1 \times 10^{11}$ |
| | C4.B | 700 C > T/700 C > T | 300 | $3.3 \times 10^8$ | $1 \times 10^{11}$ |
| | C4.C | 700 C > T/700 C > T | 300 | $3.3 \times 10^8$ | $1 \times 10^{11}$ |

*vg* viral genomes.

autofluorescence (FAF) images and as hyperreflective dots at the RPE level on corresponding sagittal spectral domain optical coherence tomography (SD-OCT) scans (Fig. 2B, Table 3 and Supplementary Fig. 4B, C). One year post-treatment, the pigmentation partially regressed in all 3 patients, leaving a hypo-fluorescent area of patchy RPE atrophy that did not progress beyond the pigmented area (Fig. 2B and Supplementary Fig. 4B,C). Another type of retinal atrophy without prior retinal pigmentation was noted in the retinotomy site and was considered procedure-related in patient C3.C and similar but smaller atrophies were also observed in patients C2.B and C4.B (Table 3 and Supplementary Fig. 5A). These retinotomy site-related areas of RPE atrophy presented on color fundus and FAF images on postoperative month 6 and persisted (without further extension) at the year 2 visit (Supplementary Fig. 5A). In patient C4.C, foveal atrophy with focal RPE and photoreceptor outer nuclear layer loss was observed in the trial eye 1-year post-treatment; the relationship of the atrophy to study drug or procedure could not be established (Table 3 and Supplementary Fig. 5B).

For all the patients, there was no significant retinal thinning observed at any time point in treated or untreated eyes, as assessed by SD-OCT (see methods for details; Supplementary Fig. 6). Best corrected visual acuity (BCVA) and low luminance visual acuity (LLVA), predominantly cone-driven assessments, are shown per patient in Supplementary Fig. 3. The mean change in BCVA from baseline was an average loss of − 4.67 (SD = 15.11) and −1.67 (SD = 4.46) letters, and for LLVA was an average gain of + 4.42 (SD = 10.77) and + 6.42 (SD = 15.20) letters, in treated and untreated eyes, respectively, at the year 1 follow-up. The larger decrease observed in BCVA in the treated group was primarily due to the severe vision loss of patient C4.C (ocular SAE), with a mean change of BCVA from baseline of − 0.45 (SD = 4.11) letters in the treated eyes when excluding this patient (Supplementary Fig. 3).

## Efficacy

The primary efficacy endpoint in this phase 1/2 clinical trial was based on the improvement of dark adaptation kinetics. This endpoint was custom-designed and assessed in a 5-year natural history study in patients with *RLBP1*-RD and was found suitable as a potential primary efficacy endpoint in an exploratory clinical trial for future validation[4]. To test the sensitivity of the cone system, light-adapted micro-perimetry with a Nidek microperimeter and visual field sensitivity with a Humphrey Field Analyzer were assessed as secondary efficacy endpoints.

Dark adaptation kinetics were measured at set times up to 6 h after exposure to a bleaching stimulus (ie. light-induced reduction of photopigment) designed to achieve a 30% bleach in healthy volunteers whose dark adaptation recovery occurs within approximately 40 min. In the *RLBP1*-RD participants in this phase 1/2 clinical trial, prior to treatment, full dark adaptation recovery was often not achieved until

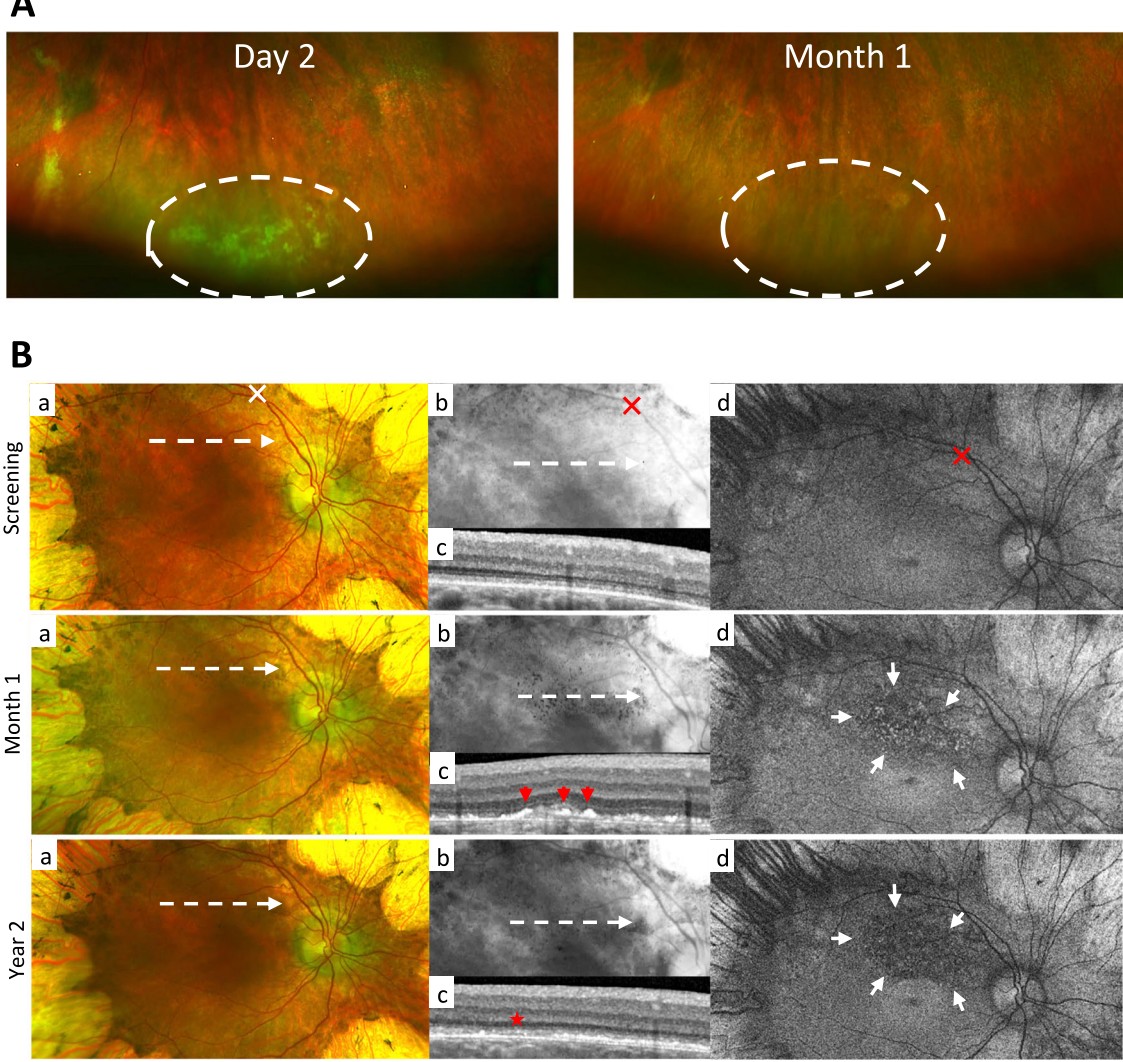

**Fig. 2 | Examples of inferior subretinal exudates and subretinal pigmented deposits. A** Inferior subretinal exudates (dashed area) in patient C2.B two days after treatment with AAV8-*RLBP1*. On day 29 post-treatment the exudates had resolved. **B** Subretinal pigmented deposits inside the superior vascular arcade shown on the color fundus image (a) and enhanced on the extracted red channel image (b) of patient C3.B 1-month post-treatment. On the corresponding SD-OCT scan (c) and FAF image (d), the deposits appear as hyperreflective dots at the RPE level (red arrows) and as hyperfluorescent dots (outlined by white arrows), respectively. Two years post-treatment, the deposits are no longer visible on either the color fundus photo or by SD-OCT. Instead, patchy loss of RPE (red star) and a hypofluorescent area are present (outlined by white arrows). The SD-OCT scan planes are shown on the fundus photos (dashed arrow). The retinotomy site of the subretinal bleb is marked (X).

6 h post-bleach, with the onset of dark-adapted recovery from the bleaching stimulus taking more than 30 min (Supplementary Fig. 7A). These results are consistent with previous results in *RLBP1*-RD participants in the natural history study[4].

To identify a treatment effect in this phase 1/2 clinical trial, pre-specified responder criteria were created from the dark-adaptation results of the natural history study (see also "Methods", Fig. 3 and Supplementary Fig. 7)[4]. Improvement was defined as any visit where post-bleach sensitivity recovery values were outside of the 75% one-sided prediction interval (based on 3 pre-treatment dark adaptation kinetics curves) at any of the pre-specified time points of 1, 2 and 3 h after bleach. A patient was considered a responder if there was improvement at a minimum of 2 or more consecutive post-treatment visits within the 1-year follow-up. Statistical analysis was conducted at the cohort level, based on responder rates at a given time point.

Of the 12 AAV8-*RLBP1*-treated eyes, there were 8, 11 and 9 responders at the 1-hour, 2-hour and 3-hour post-bleach timepoints, respectively (see definition above and Fig. 3). Of the 12 untreated fellow eyes, in contrast, there were 2, 1 and 2 responders at the 1-hour, 2-hour and 3-hour post-bleach timepoints, respectively (Fig. 3). At the cohort level, in the treated eyes, efficacy (at least 2 of 3 responders per cohort) was seen in dose cohorts C1, C2, and C4 at 1-hour post-bleach, and in all 4 cohorts at both 2 and 3-hours post-bleach. For the untreated eyes, there was no efficacy for any of the dose cohorts at any of the three timepoints (1-, 2- or 3- hours post-bleach) (Fig. 3).

Several patients with efficacy noted in the primary endpoint of dark adaptation kinetics also reported subjective improvements in activities of daily living (Supplementary Table 2). In the current trial, 11 of the 12 patients were treated in the non-dominant eye, and 1 patient was treated in the dominant eye (see Methods section). In the 11 patients treated in their non-dominant eyes, three patients (C2.B, C3.C, and C4.B) transitioned to 'no clear dominance' between their eyes (Supplementary Fig. 8), and these three patients have reported subjective improvements in their activities of daily living. These improvements were not identified using standardized quality of life questionnaires, such as the NEI visual function questionnaire-25 (VFQ-

**Table 3 | Ocular safety events**

| Cohort | ID# | Inferior sub-retinal exudates[a, h] | Vitritis[b, h] | Pigmented deposits[c, h] | Atrophy[d] |
|--------|-----|------------------------------------|----------------|--------------------------|-------------|
| C1 | C1.A | No | No | No | No |
|    | C1.B | No | No | No | No |
|    | C1.C | No | No | No | No |
| C2 | C2.A | No | No | No | No |
|    | C2.B | Yes | No | No | Yes[e, i] |
|    | C2.C | Yes | No | No | No |
| C3 | C3.A | No | Yes | No | No |
|    | C3.B | No | No | Yes | Yes[f, h] |
|    | C3.C | No | Yes | No | Yes[e, i] |
| C4 | C4.A | No | No | Yes | Yes[f, h] |
|    | C4.B | No | No | No | Yes[e, i] |
|    | C4.C | No | Yes | Yes | Yes[f, g, h] |

[a]Any inferior subretinal exudate not present at the screening/baseline visits.

[b]≥ trace of cells in the vitreous at dilated fundus examination.

[c]Pigmented deposits in the subretinal injection region were not present at the screening/baseline visits at dilated fundus examination that also presented as hyperreflective dots at the RPE level by SD-OCT.

[d]Focal atrophy detected by wide-field fundus photography and/or wide-field FAF and not present at the screening/baseline visits.

[e]Atrophy related to retinotomy site.

[f]Atrophy secondary to pigmented deposits.

[g]Foveal atrophy (relation to study-drug or procedure undecided).

[h]Considered study-drug-related.

[i]Considered procedure-related.

25) and the low luminance questionnaire (LLQ) (Supplementary Fig. 9). In the patient-reported outcomes (PROs), there were no apparent post-treatment changes beyond the level of variability for the 11 of the 12 patients treated in their non-dominant eyes. In the one patient (C2.C) treated in the dominant eye, increases of between 15 and 50 percent were noted in subsets of VFQ-25 and LLQ subscales, respectively (Supplementary Fig. 9C). There was no improvement in cone-driven secondary efficacy outcomes of central retinal sensitivity by either microperimetry or Humphrey visual field (HVF) sensitivity in the treated eye of any patient (Supplementary Fig. 10).

In this trial, three patients presented with the classic punctata albescens phenotype (Table 1). Baseline color fundus images typically displayed white puncta scattered around the vascular arcades that presented as hyperreflective subretinal deposits on the corresponding SD-OCT scans in both eyes (Fig. 4A and Supplementary Fig. 11A, B). One month after treatment, these white puncta had visibly and quantitatively regressed in all three patients only in the treated eye, whereas puncta in the untreated fellow eye remained unaltered (Fig. 4B and Supplementary Fig. 11C). Also, the extension of loss of puncta reached beyond the margin of the subretinal delivery bleb of AAV8-*RLBP1* (Fig. 4A and Supplementary Fig. 11A, B). In patients C2.C and C3.C, the reduction in puncta persisted through the 2-year follow-up whereas the puncta partially recurred after 3 months (but not to pre-treatment levels) in patient C2.B without affecting improved dark adaptation recovery after treatment (Fig. 4B).

## Discussion

In this first-in-human trial, we report safety and efficacy results on gene therapy for *RLBP1*-RD causing vision loss at an early age. Efficacy outcomes for IRD studies should be chosen carefully to reflect the functional phenotype of the specific patient population. As demonstrated for *RPE65*-RD, where multi-luminance mobility testing (MLMT) was developed as the primary endpoint during the phase-1 and phase-2 (fellow eye) clinical studies and confirmed with a validation study, which led to clinical approval[7,12]. In a 5-year natural history study, 44 *RLBP1*-RD patients' dark adaptation kinetics (among other functional

assessments) were evaluated as a novel specifically designed endpoint and hypothesized that improvement in this measure could translate into clinically meaningful improvements for this population[4].

Furthermore, a treatment area that includes both the peripherally located rods and centrally located cones would, in theory, offer maximum therapeutic benefit as demonstrated in *Rlbp1*-deficient mouse models, where AAV8-*RLBP1* vectors improved both rod and cone-mediated dark adaptation[8]. Therefore, the location chosen for the subretinal delivery (bleb) was along the retinal superior vascular arcade to allow for expansion into both rod- and cone-rich areas.

Here, 12 *RLBP1*-RD patients are successfully dosed with AAV8-*RLBP1*, albeit with considerable variation in bleb size and location. Similar variability was reported in other gene therapy studies and might be caused by differences in retinal adhesion related to the disease stage since the subretinal delivery is likely to expand towards the area with the least resistance[11,13]. Treatment with AAV8-*RLBP1* results in improved dark adaptation kinetics in all treated patients but one and further leads to subjective gains for several of the patients in other aspects of their daily life (Supplementary Table 2). Notably, despite an SAE with decreased BCVA in patient C4.C, improvement in dark adaptation recovery is still observed. Patient C3.B is the only dark adaptation non-responder, possibly due to low vector dose as a result of the formation of a smaller than expected subretinal bleb (putative vector reflux during administration) that only covered the smaller cone-rich macular area (Supplementary Fig. 2). For the remaining patients, the subretinal delivery covers rod-rich areas outside of the macula (Supplementary Fig. 2). Improved dark adaptation is observed with all vector doses, which is consistently given that the starting dose in this trial is chosen based on the minimally efficacious dose in vg/eye from *Rlbp1*-deficient mice[10].

The early foveal involvement in patients with Bothnia-type *RLBP1*-RD suggested that outcome measures based on the cone system, such as visual acuity and microperimetry, would improve also[5,14]. Albeit, dark adaptation improved across all cohorts, while none of the patients showed gains in either BCVA, LLVA, microperimetry or HVF. Longitudinal data from the natural history study showed that, in addition to dark adaptation kinetics, only HVF was potentially sensitive enough to detect a treatment effect[4]. The lack of improvement in secondary endpoints in the present trial may indicate that the disease in these patients is too advanced to demonstrate changes in cone-mediated visual functions and/or that these standard tests are not optimally designed for this disease. To rectify this, treatment at an earlier stage of this disease might be beneficial.

Visual impairment due to *RLBP1*-RD has a substantial impact on physical functioning and daily activities. The content validity of NEI VFQ-25 and the LLQ, both used in this trial, was evaluated in a dedicated study including qualitative interviews with 21 patients with *RLBP1*-RD[15]. It was found that the patients had difficulty interpreting and selecting a response for some items in the NEI VFQ-25 and LLQ, while some items were not relevant to patients' disease experience, with both gaps and overlaps in the conceptual coverage of the instruments. To adequately assess all important symptoms and associated functional impacts in *RLBP1*-RD, it was recommended to either modify one or more existing instruments or develop a new non-syndromic RP-specific instrument[15]. The results of the current treatment trial highlight how these existing PRO instruments do not show enough sensitivity to represent the patient-reported improvements in their daily activities, therefore supporting the need for developing a more sensitive PRO instrument, in agreement with the previous study in this population[15].

Sensory eye dominance occurs when the visual cortex weighs one eye's data more heavily than those of the other[16]. Sixty percent of visual function is driven by binocular vision, mediated by the dominant eye, while 40 percent of visual function is delivered monocularly, with each eye contributing equally (i.e., 20% each)[17]. Treatment of the non-

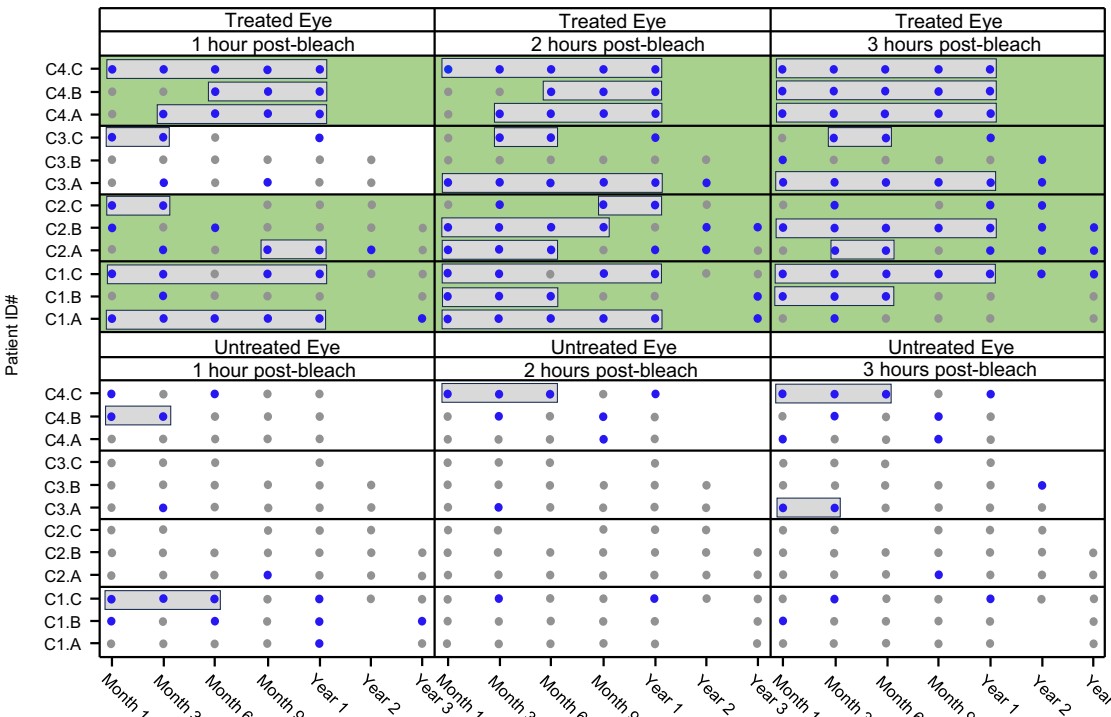

**Fig. 3 | Dark adaptation recovery kinetics.** Dark adaptation recovery for short wavelength stimulus (450 nm) outside of the pre-treatment prediction interval per timepoint, patient, visit, and cohort. Improvement (blue circles) versus non-improvement (gray circles) are shown for treated (upper panels) and untreated (lower panels) eyes at all available visits for all patients. Responders by timepoints are shown with gray boxes and efficacious cohorts by timepoints are shown with a green background. Treated eyes (*p*-values): C1 (1/2/3 h post-bleach), *p* = 0.104/ 0.08/0.104; C2 (1/2/3 h post-bleach), *p* = 0.104/0.08/0.104; C3 (1/2/3 hours post-bleach), *p* = 0.488/0.104/0.104; C4 (1/2/3 h post-bleach), *p* = 0.008/0.008/0.008. Untreated eyes (*p*-values): C1 (1/2/3 h post-bleach), *p* = 0.488/1/1; C2 (1/2/3 hours post-bleach), *p* = 1/1/1; C3 (1/2/3 h post-bleach), *p* = 1/1/0.488; C4 (1/2/3 h post-bleach), *p* = 0.488/0.488/0.488. Patient C1.A and C1.B missed the year 2 visit, and patient C2.C missed the month 6 visit due to the COVID-19 pandemic. Source data are provided as a Source Data file.

dominant eye in 11 of 12 patients could explain why there was no notable improvement in PRO subscales in the current trial.

It is well documented that after treatment with gene therapy in *RPE65*-RD patients, there is extensive cortical activation and brain plasticity, including cross-modal plasticity[18–21]. Functional magnetic resonance imaging has shown that cortical activation correlated with improvement in functional measurements such as full-field light sensitivity in *RPE65*-RD patients after treatment[19]. The current trial utilized a clinically applicable and easy to perform dominant eye test to elucidate brain plasticity. For 3 (of the 11) patients, there was a shift of eye dominance from the dominant (non-treated) eye to 'no clear dominance'. These three patients also reported subjective improvements in their activities of daily living and showed significant improvement in dark adaptation kinetics. More sensitive tests than the dominant eye test may be developed to detect more detailed improvements in integrated brain visual function in future clinical trials.

White dotlike puncta, considered a hallmark of *RLBP1*-RD, have been described to reflect mutation-related accumulation of 11-*cis*- and all-*trans*-retinyl esters in the RPE layer[22,23]. After treatment, we identify a rapid loss of white puncta on fundus images and SD-OCT scans in only the treated eye of all patients (total of three) harboring the punctata albescens phenotype, which could be explained by successful *RLBP1* coding sequence transduction, expression of CRALBP protein and restoration of the visual cycle, thus reversing retinyl ester accumulation. Importantly, the loss of puncta is not associated with any retinal thinning or RPE atrophy, during the follow-up period as assessed by imaging on sagittal SD-OCT scans and on color fundus photographs. Moreover, in two of the patients, loss of puncta persists through follow-up, indicating a long-lasting recovery of the visual cycle, whereas in patient C2.B, the puncta regresses to approximately 30% from baseline, which could indicate a diminishing treatment effect. White puncta regression is associated with improved dark adaptation recovery in all three patients and with subjective improvement in activities of their daily lives (Supplementary Table 2). This is particularly evident in patient C2.C, also the only patient receiving treatment in the dominant better-seeing eye. Furthermore, the loss of white puncta extends beyond the initial subretinal delivery area, suggesting horizontal diffusion of the viral vector in amounts that may restore CRALBP levels sufficiently. This is supported by observations in NHP where subretinal administration of AAV8 vectors has been demonstrated to diffuse extensively[24]. Our data presents the first description of a gene therapy leading to the normalization of an anatomic IRD phenotype and further suggests that the white puncta associated with *RLBP1*-RD could be a therapeutic biomarker in patients presenting this phenotype.

Subretinal AAV8-*RLBP1* treatment is overall well tolerated with dose-dependent intraocular inflammation and RPE atrophy noted as the dose-limiting toxicity concern. These AEs are not associated with any noted loss of function. Intraocular inflammation has been reported in NHPs injected with similar doses of AAV8-*RLBP1* and was also a typical finding in patients receiving gene therapy for *RPE65*-RD[7,10,13,25]. We observe asymptomatic, transient, and self-limited inferior subretinal exudates in two patients. The nature of these presumably inflammation-related deposits is unclear but similar deposits have been described in a phase 1 trial for retinitis pigmentosa GTPase regulator (*RPGR*)-associated RD that, in contrast to our trial, required treatment with an additional course of oral steroids[26]. As well, in our trial, three patients developed vitritis that was managed successfully by increasing and prolonging oral steroids. In all patients that developed signs of inflammation, either in the form of subretinal deposits or vitritis, signs, and symptoms resolved without detectable visual deterioration.

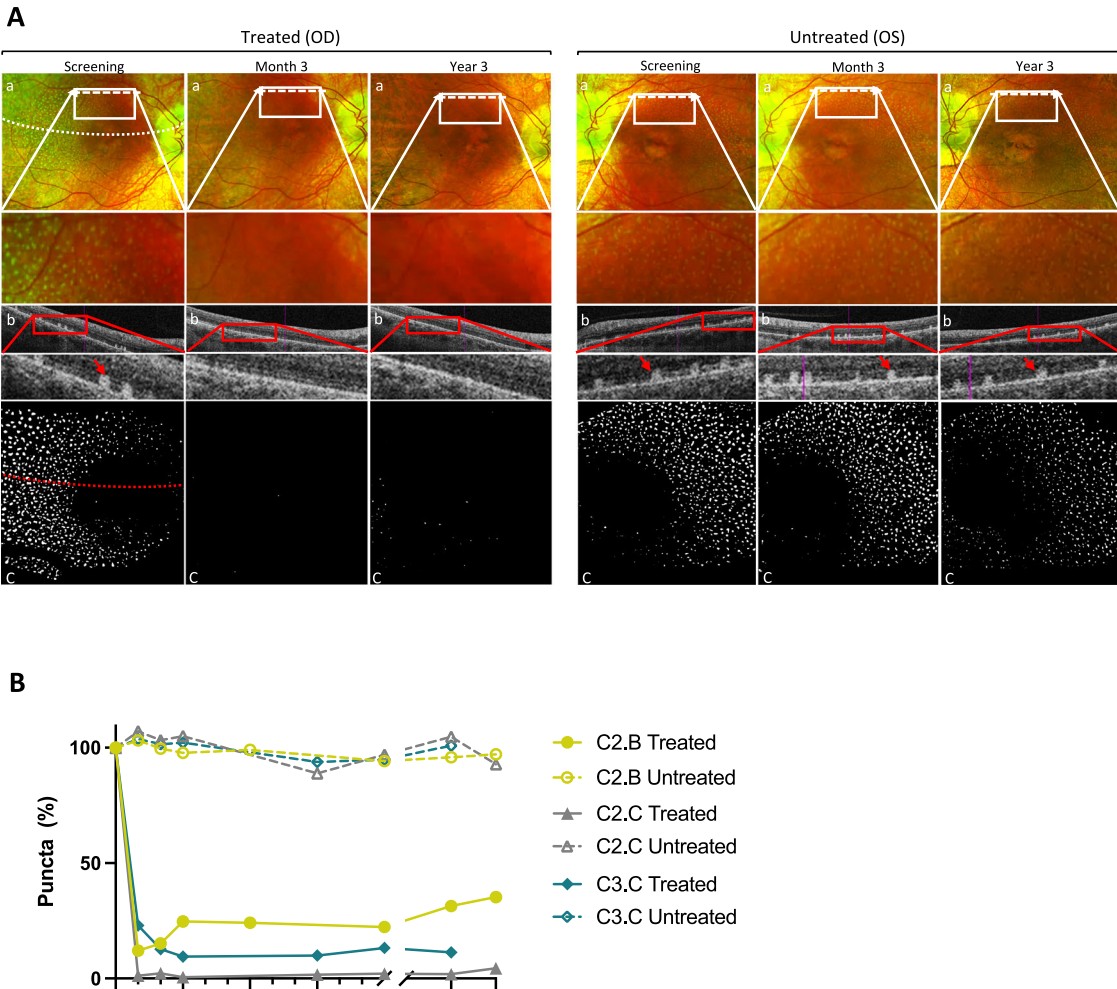

**Fig. 4 | Regression of subretinal white puncta. A** Punctata albescens deposit in patient C2.C is present in both eyes on color fundus photos (a) at the screening visit. On the corresponding SD-OCT scan (b), the puncta appear as hyperreflective dots at the RPE level (red arrows). The SD-OCT scans are shown on the color fundus photos (dashed arrows in (a)). Three months through year 3 post-treatment, the puncta persisted in the untreated eyes but were no longer visible in the treated eye. The border of the subretinal bleb is shown on the screening visit images (dotted lines in (a) and (c)). Processed threshold *en face* images (c) highlight the difference in puncta between treated and untreated eyes. **B** The change in the number of punctata albescens deposits from the screening visit to the latest visit post-treatment is shown for the treated (solid lines) and untreated eye (hatched lines) of patients C2.B (circle), C2.C (triangle) and C3.C (diamond). The quantity of puncta was captured from digitally processed fundus images within a square area of $4 \times 4$ disc diameters as described in Methods. Source data are provided as a Source Data file.

In the present trial, three distinct types of retinal atrophy are observed. Three patients in the higher AAV8-*RLBP1* dose cohorts (1 in C3 and 2 in C4) developed subretinal pigmented deposits with secondary retinal atrophy adjacent to the retinotomy site that were considered study drug-related. Hyperpigmentation after gene therapy has been described previously but, in contrast to those in the present trial, was found at the inferior margin of the subretinal bleb and associated with RPE displacement from the retinotomy site[27]. Retinal atrophy has also been described after treatment with voretigene neparvovec for *RPE65*-RD yet was not preceded by hyperpigmentation and developed progressively within 3 months post-treatment, suggesting a different causal mechanism than that observed in our trial[28–30]. The origin of the pigmented deposits is unknown but hypothesized to be due to either the clumping of pigment from aggregates of RPE cells or the infiltration of pigment-laden macrophages. In our trial, several observations suggest that the pigmented clumps originated from the RPE. First, imaging by SD-OCT positions these clumps at or directly above the RPE layer. Second, they gradually regress, leaving patchy areas of RPE atrophy. In three other patients, retinal atrophy at the retinotomy site

is observed, which is most likely procedure-related and caused by the mechanical trauma of the subretinal injection as has been described[27]. One patient who suffered severe vision loss shortly after treatment, which did not respond to oral steroids, later developed a foveal atrophy with loss of the RPE and outer nuclear layers. The initial clinical course with rapid onset in association with a history of cardiovascular pathology (high blood pressure, cardiac infarction, and pulmonary embolism) suggests that this SAE could be caused by a retinal vascular event and is considered after extensive evaluation likely not related to the viral vector.

A potential limitation of this first-in-human phase 1/2 trial is the Type 1 error rate of 10% for the hypothesis test for cohort-level responder rates, as is typically seen in first-in-human studies with small sample sizes. The difference in efficacy between the treated and non-treated eyes should, therefore, be interpreted with some caution that needs further confirmation in future studies. Another potential limitation of this trial is the inclusion of adult patients with predominantly more advanced phenotypes. Previous studies have shown that in Bothnia-type *RLBP1*-RD, macular function, and structure are affected

already in childhood and that visual acuity declines from early adulthood[3,5].

In future studies, we would propose treatment of younger patients where visual function is less affected, and the macula and retina are better preserved. Eligibility may also be based on OCT-based anatomical indicators such as ellipsoid line width and central retinal thickness, as recently proposed[1]. In addition, the structure-function correlation observed between the regression of white puncta and improvement in dark adaptation kinetics supports the treatment of patients with a punctata albescens phenotype. Finally, we suggest using the vector doses in C1 or C2 ($5 \times 10^9$ and $1 \times 10^{10}$ vg/eye, respectively) that have indicative efficacy yet limited risk of inducing inflammation or atrophy.

In conclusion, in our first-in-human phase 1/2 gene therapy trial for *RLBP1*-RD, the drug is overall well tolerated, and drug-related toxicity was mainly observed at the higher viral vector doses. Treatment with AAV8-*RLBP1* at all doses improves dark adaptation recovery and adaptation to vision in low luminance and regresses the anatomic disease phenotype.

## Methods
### Details of trial protocol
**Ethical and regulatory approval.** This trial (https://clinicaltrials.gov/ct2/show/NCT03374657, https://www.clinicaltrialsregister.eu/ctr-search/trial/2016-002696-10/SE, first submitted on Dec 11, 2017) was sponsored by Novartis Pharma AG and performed with approval from the Swedish National Ethics Committee (Etikprövningsmyndigheten, EPM) and the Swedish Medical Products Agency (Läkemedelsverket, LV), and adhered with all applicable laws and regulations including the International Conference on Harmonization Guidelines of Good Clinical Practice and the Declaration of Helsinki.

**Trial design.** This was a prospective, interventional, non-randomized, non-confirmatory, open-label single ascending dose, gene replacement-therapy trial to assess the safety, tolerability, and efficacy of AAV8-*RLBP1* vector in patients with biallelic loss-of-function mutations in the *RLBP1* gene. The trial was conducted at St. Erik Eye Hospital in Stockholm, Sweden, with written informed consent provided by all patients. Patients with Bothnia-type *RLBP1*-RD were enrolled between August 22, 2018, and March 10, 2021, and patient data was analyzed between August 2018 and June 2022. Enrolled patients were only compensated for travel and accommodation expenses. The first-in-human phase 1/2 trial used a staggered patient enrollment with continuous data reviews to limit as much unforeseen risk as possible prior to enrolling each patient in each cohort or initiating another cohort. As is standard in first-in-human ocular studies, only one eye (designated as the treated eye) was dosed per patient. Following two screening visits, patients had a baseline evaluation followed by the surgical visit (day 1). All baseline safety and efficacy evaluation results had to be available prior to dosing. Prior to surgery, the study eye was determined based on the inclusion/exclusion criteria and eye dominance. Subsequent outpatient visits are scheduled after surgery at days 2, 3, and 4, with an additional visit between days 5 and 9 and at weeks 2, 3, at months 1, 2, 3, 6, and 9; and at years 1, 2, 3 and will continue to years 4 and 5.

Prior to dosing each patient in a cohort (i.e., 2nd and 3rd patient in each cohort), a safety team consisting of the site principal investigator, sponsor medical expert, and sponsor statistician reviewed all safety data for every available patient. These safety meetings included at least one month of data from the most recently dosed patient. A data monitoring committee (DMC) was chartered for an independent, unblinded safety review. Prior to dose escalation, the DMC performed an interim review of all available safety and tolerability data (e.g., AEs, SAEs, ophthalmic exams including SD-OCT and color fundus photographs, vital signs, and clinical laboratory assessments) as well as of the primary efficacy data. These interim analyses included at least 3 months of data from the final patient of the previous cohort.

**Key Inclusion and exclusion criteria.** The trial population consisted of patients who passed screening assessments, complied with inclusion/exclusion criteria, and provided written consent.

Key inclusion criteria were:
1. Male and female patients aged 18 to 70 years inclusive. All female patients were required to have negative pregnancy test results.
2. Best corrected visual acuity (BCVA) in the trial eye at the screening 1 visit ≤ 60 Early Treatment Diabetic Retinopathy Study (ETDRS) letters. In C1, patients with BCVA from 35 ETDRS letters to hand motion visual acuity were included. For C2 and beyond, patients with BCVA of 60 ETDRS letters to hand motion were enrolled. Consideration for inclusion in C1 and C2 was only given to patients with visual acuity as low as Light Perception at the screening 1 visit, if they met all other inclusion/exclusion criteria and at the agreement of the sponsor and principal investigators.
3. Clinical diagnosis of Bothnia dystrophy, Newfoundland rod-cone dystrophy, or other progressive retinitis pigmentosa phenotype with biallelic mutations in the *RLBP1* gene verified by Clinical Laboratory Improvement Amendments (CLIA) genetic testing.
4. Visible photoreceptor (outer nuclear) and RPE layers on standard SD-OCT scan in the study eye at the screening 1 visit, as confirmed by the central reading center.
5. Dark adaptation bleaching effect in the study eye with short wavelength stimulus of > 1.0 log unit at 2 of 3 measurements obtained prior to treatment.

Key exclusion criteria were:
1. Pre-existing eye conditions that would preclude the planned surgery or interfere with the interpretation of trial endpoints.
2. Any active infection, intraocular inflammation, or ocular disease involving ocular adnexa in either eye.
3. Pre-existing eye conditions that would preclude the planned surgery or interfere with the interpretation of trial endpoints; for example: glaucoma (IOP ≥ 25 mmHg despite treatment with anti-glaucoma medication, or low tension glaucoma), corneal or significant lenticular opacities, retinal vascular occlusion, retinal detachment, macular hole, or choroidal neovascularization of any cause.
4. Current treatment for active systemic infection.
5. Complicating systemic diseases or clinically significant abnormal laboratory values.
6. Women who were pregnant, lactating, or women of childbearing potential were defined as all women physiologically capable of becoming pregnant unless they were using highly effective methods of contraception during dosing and for two months after treatment.

**Dose rationale.** The AAV8-*RLBP1* vector was designed to deliver the *RLBP1* gene to the RPE and Müller cells, where the gene product, CRALBP, is normally present (Supplementary Fig. 1)[8,10]. The safety findings that determined the no-observed-adverse-effect level (NOAEL) concentration of AAV8-*RLBP1* in an NHP toxicology study were intraocular inflammation and retinal thinning[10]. The NOAEL concentration ($3.3 \times 10^9$ vg/eye) in NHP was essentially the same as the minimal efficacious dose in *Rlbp1*-deficient mice and was consequentially chosen for the first cohort of the trial (Table 2)[8]. Based on half-log concentration escalation, the dose of $5 \times 10^9$ vg/eye was delivered in a 150 μl volume in C1 and was determined safe, in C2, keeping the same vector concentration as in C1, the volume was increased to 300 μL. This resulted in a total dose of $1 \times 10^{10}$ vg/eye. For subsequent cohorts, 300 μL was administered and the concentrations

escalated to $1 \times 10^8$ vg/μL for C3 (total dose of $3 \times 10^{10}$ vg/eye) and $3.3 \times 10^8$ vg/μL for C4 (total dose of $1 \times 10^{11}$ vg/eye).

## Prophylactic steroid treatment

A preventive peri-operative regimen of oral and topical steroids, similar to the regimen used for voretigene nepavornec gene therapy trial, was used[7]. This regimen included 40 mg prednisolone daily for 8 days (starting 3 days prior to AAV8-*RLBP1* dosing), tapering to 20 mg daily for 5 days, and then to 20 mg every other day for 5 days. In addition, topical dexamethasone was administered 3 times daily for 2 weeks, tapered to 2 times daily for 1 week, and 1 time daily for 1 week. A modified, prolonged prednisolone regimen, as recommended by the DMC for C4, consisted of 40 mg daily for 4 weeks, tapering to 30 mg daily for 2 weeks, to 20 mg daily for 2 weeks, to 10 mg daily for 2 weeks and 5 mg daily for 2 weeks.

## Surgical procedure

Surgeries were performed under general anesthesia using a standard 3-port 23-gauge pars plana vitrectomy procedure. A core vitrectomy was performed including confirmation of a complete posterior vitreous detachment. Using a subretinal injector (D.O.R.C. International BV, Zuidland, The Netherlands) loaded with AAV8-*RLBP1* vector, the cannula tip was placed temporally to the optic nerve and in the vicinity of the vascular arcades. Light pressure was applied to the tip to create a "whitening" of the retina, whereafter the assistant injected the AAV8-*RLBP1* vector to create a subretinal bleb. The subretinal injection cannula was removed and a fluid-air exchange was performed to remove any test drug material from inadvertently leaking into the vitreous. All sclerotomies were sutured. Supine head position was maintained for 4–6 h, by which time the subretinal bleb had resorbed.

## Database management and quality control

Data, data queries, and AEs were entered into a digital case report form (CRF) at the site and then monitored by a Novartis Clinical Research Associate and reviewed regularly by medical experts. Some data were sent to external vendors for quality control. A central reading center reviewed and analyzed (where required) all SD-OCTs, microperimetry, and dark adaptation data. Laboratory samples were sent to the central laboratory (Clinical Reference Laboratory (CRL), Lenexa, Kansas, replaced by Q² Solutions, Durham, NC), and were either analyzed by the central lab or were sent elsewhere for further analysis and/or storage (Lab Corp Development, Burlington, NC).

## Masking

This phase 1/2 trial was partially masked. The participants were not masked. The treating physicians and personnel at the surgical location (surgeons, anesthesiologist, operating room personnel and others) were not masked. At the clinical site, there were masked ophthalmologists. The remaining assessors at the clinical site (ophthalmologist, study nurse, ophthalmic technician, etc.) performing the ophthalmic examinations were masked to the study (treated) eye.

## Safety outcomes

The pre-specified primary safety endpoint of this first-in-human trial was the safety and tolerability of AAV8-*RLBP1*. The following safety parameters were descriptively analyzed to assess treatment-emergent changes: AEs and SAEs, change from baseline in vital signs, clinical laboratory evaluations, BCVA (ETDRS letters), intraocular pressure, slit lamp examination of the anterior and posterior segments, dilated fundus examination, SD-OCT, and color fundus photography. Trial-stopping rules included SAEs related to the study drug in two or more patients, or hypersensitivity to the drug. Cohort-stopping rules included loss of BCVA ($\geq$ 30 letters), degree of ocular inflammation (persistent grade 2 anterior chamber inflammation and/or grade 3 vitreous haze), or degree of retinal thinning ($\geq$ 40%) in 2 or more patients per cohort.

**Visual acuity.** BCVA (standard luminance conditions) and low luminance visual acuity (LLVA) were assessed on both eyes using ETDRS visual acuity charts at 4 m or 1 m (for patients that could not read the 4 m chart) and converted to LogMAR values[31]. LLVA was assessed with the use of a 2.0 log-unit neutral density filter placed into a study lens holder for the tested eye.

**Spectral domain-optical coherence tomography.** For safety monitoring, spectral domain-optical coherence tomography (SD-OCT) was performed on both eyes to generate retinal thickness maps of 9 central subfields using the built-in software functionality (Heidelberg Eye Explorer version 1.10.4.0, Spectralis, Heidelberg Engineering, GmbH). At screening for each patient, 49 B-scans centered on the macula (20° x 20°, ART set to 16 and high-resolution mode) were captured. Depending on the fixation of the patient, scan density was adjusted to 97 B-scans or 25 B-scans. At screening, 9 subfield maps were also captured using the Cirrus SD-OCT (Zeiss, Carl Zeiss AB, Stockholm, Sweden). For some patients, insufficient fixation made it impossible to use the Spectralis, and subfield maps were instead captured using the Cirrus SD-OCT. At follow-up visits, subfield maps were captured using the same setting and equipment as used at the screening visit.

**Ultra-widefield (UWF) multicolor scanning laser ophthalmoscopy and fundus autofluorescence (FAF).** Two-hundred degree multicolor (red - 633 nm, green - 532 nm) images were captured using the Optos UWF scanning laser ophthalmoscopy device (Optos Advance ver 4.4R2, Optos Inc, Marlborough, MA). Fundus autofluorescence images were generated using the devices green wavelength for excitation.

## Efficacy outcomes

**Full-field dark-adaptation test.** The pre-specified primary efficacy end-point was dark adaptation kinetics and recovery after treatment with AAV8-*RLBP1*. Dark adaptation is the adjustment of the eye to low light intensities over time. The dark adaptation test of retinal sensitivity used in this trial was specifically designed for *RLBP1*-RD patients and differed from the usual white-light dark-adapted full-field sensitivity threshold (FST) test typically assessed in patients with retinal degenerations[4,32]. Dark-adapted retinal sensitivity was measured after the eyes were dark-adapted overnight using an eye patch to block light from entering the eye and the pupils were dilated. The Diagnosys Espion system with the ColorDome™ LED stimulator (Diagnosys Colordome Software Version V6.64.15, Diagnosys LLC, Lowell, MA) was used for testing. Thresholds were measured to full-field stimuli of either short (450 nm) or long (632 nm) wavelength, which were randomly interleaved. To assess this, the lowest light level (threshold) a patient could just detect in a dark room was measured, giving a baseline measurement (pre-bleach). Next, the patient was exposed to a steady light source, creating a 30% bleach in healthy volunteers. At different time points after bleaching (0, 7.5, 15, 30 min, 1, 2, 3, 6 hs) the lowest level of light that the patient could detect with that eye was recorded, to create a dark adaptation curve. A measure of the time course of dark adaptation using this test developed in-house test showed that while healthy volunteers could dark adapt in 20–30 min, it typically took patients with *RLBP1*-RD up to 6 h to recover to their baseline values[4]. Using three pre-treatment tests (performed at 2 screening visits and at baseline), a prediction interval was created for each eye at each timepoint. The goal was to measure any change from baseline by seeing how much a patient's sensitivity fell outside their previously calculated prediction interval estimated from pre-treatment dark adaptation testing. The top of the range created by a 1-sided 75% prediction interval with 2 degrees of freedom was used as the cutoff for improvement. This cutoff is equal to the top cut-off of the 2-sided 50% prediction interval (Supplementary Fig. 7A). The 1-, 2- and 3-h post bleach timepoints were selected as clinically relevant for the evaluation of a treatment effect. Improvement was defined as any

visit where post-bleach sensitivity recovery values were outside of the 75% one-sided prediction interval (based on pre-treatment kinetics curves) at any of the pre-specified time points of 1, 2, and 3 h after bleach. Using an interval of this size implies that with no change from treatment, there is still a 25% chance that a given observation will fall above the top range. Or, conversely, there is a 25% probability of getting a false positive improvement. A patient was considered a responder at a specific timepoint post-beach if there was improvement at a minimum of 2 or more consecutive post-treatment visits within the 1-year follow-up. As there are five post-treatment visits (months 1, 3, 6, 9, and 12) to potentially show improvement, the false positive rate for a responder is 20%. Statistical analysis was conducted at the cohort level, based on responder rates. The false positive rate for a responder was the value used in our hypothesis testing to evaluate within each cohort if the responder rate was greater than what could have occurred by chance (20%). The alpha level is also set to 0.1 instead of 0.05, which is typically used with small sample sizes, as in the current trial.

Supplementary Fig. 7A shows a pre-treatment dark adaptation curve, this is the average of the 3 pre-treatment values with the prediction interval. Once treated with AAV8-*RLBP1*, patients' dark adaptation was tested at the same time points at every visit. All curves were plotted together so each visit could be compared to the pre-treatment visits. As an example, Supplementary Fig. 7B shows dark adaptation curves for all the visits of patient C1.A. To summarize all patients and eyes across time points, a dot plot was created (Supplementary Fig. 7C). This was done by converting every visit into one line. If the patient took the dark adaptation test at that visit a circle was put into the plot. If the sensitivity value at a given time point for a specific visit was outside the prediction interval, this is shown as a filled blue circle. A gray-filled circle represents values within the prediction interval (Fig. 3).

**Light-adapted microperimetry.** Light-adapted microperimetry or fundus-controlled microperimetry was assessed (Nidek MP-3 ver 1.1.2.00, Nidek CO Ltd, Aishi, Japan) to generate a sensitivity map of the retina by measuring the patient's ability or inability to perceive light stimuli of different intensities projected at different locations on the retina. A sensitivity map adjusted for fixation stability was generated by observing a live infrared image of the retina. Each specific retinal location was associated with a value of the brightness of the stimulus seen by the patient. The data from microperimetry was then registered onto a digital color fundus image. The points from one eye at one visit were ordered from the most sensitive to the least and the ranks and their intensities were graphed to create a curve for each eye. From this curve, the area under the curve (AUC) could be calculated. The greater the AUC, the better the patient responded to the stimuli.

**Humphrey visual field test (HVF).** The Swedish Interactive Threshold Algorithm (SITA standard) 30-2 program with stimulus size III was used for visual field testing (Humphrey Field Analyzer 2i, Zeiss, Carl Zeiss AB, Stockholm, Sweden). Each eye was tested separately, and the non-tested eye was patched. Performance (mean deviation) and compliance measures (number of fixation losses, false positive errors, and false negative errors) were recorded for each eye in the electronic case report form (eCRF).

**Dominant eye test.** Patients were seated facing an eye chart for measuring acuity and asked to stretch out both arms, creating a small aperture with the thumbs and index fingers of both hands. With both eyes open, they were asked to fixate on a letter on the chart through the aperture and then, while keeping both eyes open, to bring their hands slowly toward their face. The eye that the hands were drawn toward was assigned the dominant eye. As an alternative approach, a card with a small hole in the center could also be used instead of an aperture made with the hand. The patient could do the test more than once and with different methods if needed, but eventually, the assessor assigned a single answer, which was one of the following: Right Eye Dominant, Left eye Dominant, or No clear dominance.

**Patient reported outcomes (PROs).** The National Eye Institute Visual Functioning Questionnaire-25 (VFQ-25) and Low Luminance Questionnaire (LLQ) PROs were administered[33,34]. In addition, during each visit, the patients were also asked about their experiences in the trial and were encouraged to describe their functional vision and to give specific examples of any daily-life changes pre- and post-treatment.

### Exploratory assessments
**Quantification of punctata albescens deposits.** The number of punctata was evaluated from the fundus images at screening and follow up visits in punctata albescens patients (ImageJ ver 1.54b, https://imagej.net/ij/[35],). An area of 4 x 4 disc diameters (horizontal: from foveal center to 3 disc diameters temporally and 1 disc diameter nasally, vertical: from the foveal center to 2 disc diameters superiorly and inferiorly) was cropped and the green channel was extracted (Supplementary Fig. 10C). The following processing was done to highlight the punctata: a rolling ball background subtraction was performed to correct for the uneven illuminated background and then contrast enhancement was performed using adaptive histogram equalization. A bandpass filter was applied to enhance the punctata. Supplementary Fig. 10C shows an example image processed with rolling ball background subtraction, enhanced local contrast, and bandpass filter plugins from ImageJ. The number of punctata was counted automatically after applying Otsu thresholding to locate high-intensity regions in the processed image (GraphPad ver 9, GraphPad Software, Boston, MA) (Fig. 4B).

### Statistics
**Analysis of the primary efficacy variable.** Efficacy analyses related to the primary objective were based on the evaluation of sensitivity recovery at 1, 2, and 3 hs post-bleach, as well as other dark adaptation endpoints as described below. Sensitivity recovery (SR) was defined as follows:

$$SR(t, V_i) = STV(t_{pre}, V_i) - STV(t, V_i),$$

where $STV(t, V_i)$ is the sensitivity threshold value (in $\log_{10}$ scale) at post-bleach time point t, at the i[th] visit $V_i$; and $t_{pre}$ is the pre-bleach time point (following overnight dark adaptation, which patients are assumed to have reached their maximum sensitivity). At each $V_i$ and each t, the $STV(t, V_i)$ value is derived as the median of three replicate measurements.

Dark adaptation variables related to the primary efficacy objective include:
- Sensitivity recovery at 1 h post-bleach: SR(1 h, $V_i$) (primary analysis)
- Sensitivity recovery at 2 hs post-bleach: SR(2 h, $V_i$) (primary analysis)
- Sensitivity recovery at 3 hs post-bleach: SR(3 h, $V_i$) (primary analysis)

A responder analysis was performed for the SR(t)s listed above for each cohort as the primary analysis. A patient is considered a responder if ≥2 consecutive visits SR(t, $V_i$) values are observed to be outside of the patient's prediction interval within one year after treatment. A patient-specific one-sided 75% prediction interval (PI) was established using multiple baseline DA values, based on a Student's t distribution with B-1 degrees of freedom (B is the number of baseline dark adaptation sessions).

It can be shown that the chance of observing at least 2 consecutive false positive responses (i.e., sensitivity recovery falling outside of the 75% PI by chance) out of 5 post treatment visits is 20% (based on a binomial distribution bin(n,p) with $n = 5$ and $p = 0.25$). The responder rate is therefore set to 20%, and at the cohort level, the hypothesis related to the primary analysis for each SR(t) are:

$H_0$: The responder rate is $\leq 20\%$

$H_1$: The responder rate is $> 20\%$

An exact binomial test is used to test the above hypothesis with a one-sided alpha level of 0.1 (SAS ver 9.4, SAS Institute Inc, Cary, NC). Due to the discreteness of the test the minimal number of responders to reject the null hypothesis can be determined. For example, 2 patients out of 3 in a cohort lead to the rejection of the null hypothesis with an alpha of 0.1. As this exact test is discrete, the alpha level must be slightly adjusted upwards (alpha = 0.104).

### Reporting summary

Further information on research design is available in the Nature Portfolio Reporting Summary linked to this article.

## Data availability

Due to the small sample size and patient privacy restrictions, it is unlikely that individual patient data can be shared. Beginning 3 months and ending 24 months after article publication, all requests for raw and analyzed data and materials should be made to the corresponding author and are reviewed within a month by the St. Erik Eye Hospital and the sponsor, Novartis Pharma AG, to verify whether the request is subject to any intellectual property or confidentiality obligations or patient privacy issues. Patient-related data not included in the paper were generated as part of the clinical trial and may be subject to patient confidentiality. Any data and materials that can be shared will be released via a material transfer agreement by investigators to achieve the aims of the approved proposal. The trial protocol is available from the corresponding author upon reasonable request. Source data are provided in this paper.

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

## Acknowledgements
We thank all trial participants for their commitment to the trial and for attending their follow-up visits, and staff members of the St. Erik Eye Hospital for their support throughout the trial. We thank the Duke Reading Center, Duke Eye Center, Durham, NC, USA. We thank Ted Dryja, MD, and Vivian Choi, Ph.D (ex-Novartis employees) for their key roles in the initiation of this gene therapy project. The trial was sponsored by Novartis Pharma AG (https://clinicaltrials.gov/study/NCT03374657).

## Author contributions
A.K., M.B., S.W., and A.G. recruited and monitored trial participants during screening and follow-up visits. A.K. performed the gene therapy surgery. A.G. assisted with the gene therapy surgery. A.K., N.R., K.H., C.W., A.V., S.W., and A.G. collected data and performed the data analysis. C.B. performed preclinical studies to validate the vector. H.A. was the biological safety officer. A.K., K.H., and K.S. designed the trial protocol. A.K., N.R., K.H., C.W., N.J., C.G., M.L., C.B., A.G., and H.A. wrote and revised the manuscript. All the authors provided scientific input and read and approved the manuscript.

## Funding

## Competing interests
N.R., K.H., C.W., N.J., C.B., K.S., M.L., and C.G. were employees of the sponsor of the trial, Novartis Pharma AG. The remaining authors declare no competing interests.
