## [Peer Review File · Nature Communications]

Editorial Note: This manuscript has been previously reviewed at another journal that is not operating a transparent peer review scheme. This document only contains reviewer comments and rebuttal letters for versions considered at Nature Communications

REVIEWERS' COMMENTS

Reviewer #1 (Remarks to the Author):

I have no additional comments.

Reviewer #2 (Remarks to the Author):

The authors have adequately addressed the points raised, putting the study and its results in perspective and acknowledging the limitations.

Reviewer #3 (Remarks to the Author):

My previous comments are still relevant and now updated. The authors have addressed all of the concerns. No further question or concerns.

1) The authors report on subretinal delivery of AAV RLBP1 in a dose escalation trial in 12 patients with Retinitis Punctata Albescens (RPA) an autosomal recessive disorder, with 3 years of followup. Patients varied from those with mild disease with moderate vision loss to those with advanced stages, with widespread atrophy and poor vision. Safety and tolerability were assessed, and both dose-dependent intraocular inflammation and RPE atrophy were noted. Primary efficacy was assessed via changes in dark adaptation and reported to be positive.

2) The authors have previously published a 5-year natural history study to explore the novel primary endpoint of change in dark adaptation.

3) Gene-based therapies have been used successfully to treat RPE-65 associated LCA using subretinal delivery of an AAV vector-based gene therapy. The current study would extend treatment to another autosomal recessive RP and would be the first therapy for this form of RP.

Reviewer #4 (Remarks to the Author):

This submission proposes to evaluate the safety and efficacy of a single unilateral sub-retinal injection of an AAV8-RLPB1 gene therapy construct to treat bi-allelic mutation in the RLPB1 gene. My prior concerns related to the formulation, rationale and support for the reporter function, primarily based upon an extended dark adaptation endpoint, have been addressed and edits made to the text. These changes have largely addressed my concerns. The addition of statistical clarity regarding the meaning of multiple sequential dark adaptational improvements aided reader understanding. Additional clarification of complication relationship to intervention, management of complications and future concerns of dosage are appreciated.

Point-by-point reply: NCOMMS-24-11049-T

Reviewer #1 (Remarks to the Author):

I have no additional comments.

Author reply: no action taken

Reviewer #2 (Remarks to the Author):

The authors have adequately addressed the points raised, putting the study and its results in perspective and acknowledging the limitations.

Author reply: we thank the reviewer for the supportive comment. No action taken

Reviewer #3 (Remarks to the Author):

My previous comments are still relevant and now updated. The authors have addressed all of the concerns. No further question or concerns.

1) The authors report on subretinal delivery of AAV RLBP1 in a dose escalation trial in 12 patients with Retinitis Punctata Albescens (RPA) an autosomal recessive disorder, with 3 years of followup. Patients varied from those with mild disease with moderate vision loss to those with advanced stages, with widespread atrophy and poor vision. Safety and tolerability were assessed, and both dose-dependent intraocular inflammation and RPE atrophy were noted. Primary efficacy was assessed via changes in dark adaptation and reported to be positive.

2) The authors have previously published a 5-year natural history study to explore the novel primary endpoint of change in dark adaptation.

3) Gene-based therapies have been used successfully to treat RPE-65 associated LCA using subretinal delivery of an AAV vector-based gene therapy. The current study would extend treatment to another autosomal recessive RP and would be the first therapy for this form of RP.

Author reply: we thank the reviewer for the supportive comments. No action taken

Reviewer #4 (Remarks to the Author):

This submission proposes to evaluate the safety and efficacy of a single unilateral sub-retinal injection of an AAV8-RLPB1 gene therapy construct to treat bi-allelic mutation in the RLPB1 gene. My prior concerns related to the formulation, rationale and support for the reporter function, primarily based upon an extended dark adaptation endpoint, have been addressed and edits made to the text. These changes have largely addressed my concerns. The addition of statistical clarity regarding the meaning of multiple sequential dark adaptational improvements aided reader understanding. Additional clarification of complication relationship to intervention, management of complications and future concerns of dosage are appreciated.

Author reply: we thank the reviewer for the supportive comments. No action taken